# Classifying Upper Arm Gym-Workouts via Convolutional Neural Network by Imputing a Biopotential-Kinematic Relationship

Ji-Hyeon Yoo [1], Ho-Jin Jung [1], Yi-Sue Jung [1], Yoon-Bee Kim [1], Chang-Jae Lee [2], Sung-Tae Shin [3,*] and Han-Ul Yoon [1,2,*,†]

1. Division of Computer and Telecommunication Engineering, Yonsei University, Wonju 26493, Korea; starsand96@yonsei.ac.kr (J.-H.Y.); wo8672@yonsei.ac.kr (H.-J.J.); yisue@yonsei.ac.kr (Y.-S.J.); yoonbee7@yonsei.ac.kr (Y.-B.K.)
2. Department of Computer Science, Yonsei University, Wonju 26493, Korea; cjlee7128@yonsei.ac.kr
3. Department of Mechanical Engineering, Dong-A University, Busan 49315, Korea
* Correspondence: stshin@dau.ac.kr (S.-T.S.); huyoon@yonsei.ac.kr (H.-U.Y.); Tel.: +82-51-200-7653 (S.-T.S.); +82-33-760-2235 (H.-U.Y.)
† Current address: 1 Yonseidae-gil, Wonju 26493, Korea.

**Abstract:** This paper proposes a systemic approach to upper arm gym-workout classification according to spatio-temporal features depicted by biopotential as well as joint kinematics. The key idea of the proposed approach is to impute a biopotential-kinematic relationship by merging the joint kinematic data into a multichannel electromyography signal and visualizing the merged biopotential-kinematic data as an image. Under this approach, the biopotential-kinematic relationship can be imputed by counting on the functionality of a convolutional neural network: an automatic feature extractor followed by a classifier. First, while a professional trainer is demonstrating upper arm gym-workouts, electromyography and joint kinematic data are measured by an armband-type surface electromyography (sEMG) sensor and a RGB-d camera, respectively. Next, the measured data are augmented by adopting the amplitude adjusted Fourier Transform. Then, the augmented electromyography and joint kinematic data are visualized as one image by merging and calculating pixel components in three different ways. Lastly, for each visualized image type, upper arm gym-workout classification is performed via the convolutional neural network. To analyze classification accuracy, two-way rANOVA is performed with two factors: the level of data augmentation and visualized image type. The classification result substantiates that a biopotential-kinematic relationship can be successfully imputed by merging joint kinematic data in-between biceps- and triceps-electromyography channels and visualizing as a time-series heatmap image.

**Keywords:** imputing a biopotential-kinematic relationship; kinematic and biopotential data analysis; human behavior classification; upper arm gym-workout classification; convolutional neural network

## 1. Introduction

Nowadays, many people regularly perform gym-workouts to prevent enervation and invigorate the activities of daily living. The gym-workout protocol has been encouraged for both promoting an individual fitness and expediting patient's rehabilitation process [1–3]. Prerequisite for maximizing the efficacy of exercise and preventing unexpected injury is that a person must work out with a correct posture as well as target muscle stimulation while performing exercises such as arm-curl, dead-lift, kettle-bell squat, and so on [1,4]. Habitually failing to fulfill either the former or the latter prerequisite may cause serious injuries on muscular-tendon or musculo-skeletal mechanism [5,6]. Accordingly, people's interests about exercise monitoring systems have been grown as the number of both gym-goers and home trainees increase.

Computer vision-based approach is one main pillar to build exercise monitoring systems. In the computer vision-based approaches, the exercise monitoring system is typically equipped with RGB-d (or RGB) cameras such as Microsoft Kinect, Intel Realsense, etc. [7,8]. For the captured video frame or image, people's joint positions are detected and estimated with the support of deep learning-based algorithms, then their posture is displayed as a skeleton [9,10]. Torres et al. proposed an upper limb strength training system in which a user's posture was detected by Kinect v2 [11]. Nagarkoti et al. presented a mobile phone video recording-based approach to realtime indoor workout analysis [12] and a similar approach was reported by Liu and Chu [13]. The aforementioned studies in [1–3] are of computer vision-based approaches as well; specifically, exercise posture correction for home trainee [1], posture modeling [2], and posture correction during rehabilitation training [3].

Surface electromyography (sEMG)-based approach is the other main pillar which enables to monitor body posture and movements. Due to the inter-subject variability (or difference) of the signal, the sEMG-based approach is used to be supported by advanced neural networks; especially, since sEMG is multichannel time series data, recurrent neural network (RNN), long short-term memory (LSTM), and deep belief network (DBN) are mostly used for desired classification [14–16]. Quivira et al. proposed an approach to classify dynamic hand motions by translating sEMG signals via RNN [17]. Orjuela-Cañón et al. employed deep neural network (DNN) architecture to solve the classification problem of wrist position based on sEMG data [18]. The more dexterous motion is, the more advantageous information sEMG carries; accordingly, similar approaches for the classification of finger, hand and arm movements were introduced in [19–21]. In addition, the monitored muscle activation as well as generated force can directly serve as an indicator to tell whether the target muscle is stimulated during workouts. The sEMG-based force monitoring approaches can be categorized as follows: neural network (NN)-based approach [22–26], dynamic-model and software-based approach [27–29], optimization-based approach [30,31].

The above introduced approaches, so far now, have solely used either computer-vision or sEMG. In contrast, we know that it would be beneficial to endow exercise monitoring systems with multimodality in a sense of information gathering. Kim et al. proposed a posture monitoring system in which both sEMG and inertial measurement unit (IMU) were employed to estimate human motion [32] and Xu et al. introduced a similar approach [33]. Wang used IMU together with Apple Watch to reconstruct arm posture for upper-body exercises [34]. Several studies utilized both sEMG sensor and computer-vision; for instance, equinus foot treatment system by Araújo et al. [35], prosthesis control interface by Blana et al. [36], and rehabilitation video games by Rincon et al. [37] and Esfahlani et al. [38], respectively. As aforementioned, "an individual is performing a proper and exact workout" means that his or her joints are following a correct posture and target muscles are being stimulated primarily. Even though there have been existing studies, an exercise monitoring approach in perspectives of muscle physiology as well as joint kinematics has not yet been fully considered.

Studies have striven for demystifying a relationship between sEMG activation and joint kinematics. Michieletto et al. and Triwiyanto et al. proposed Gaussian–Markov process-based elbow joint angle estimation techniques, respectively [39,40]. Initiated from Hill's muscle model, Han et al. introduced a state space model for elbow joint and Zeng et al. presented their works for knee joint [41,42]. Pradhan et al. used VICON-based 3D motion capture and sEMG data and reported that a relationship between those two were still fuzzy [43]. As seen in the case of sEMG-based force monitoring above, NN-based approach might be a reasonable solution to estimate elbow joint kinematics from sEMG signals [44–47]. Namely, due to the characteristics of sEMG signal (including individual differences), a relationship between sEMG activation and joint kinematics not so straightforward and existing findings are still debatable [48,49].

To address the issues above, we propose a systemic approach to upper arm gym-workout classification via convolutional neural network (CNN). The proposed approach can be regarded as a subsolution of the larger class of problem in the design of exercise monitoring system with a specific example of upper arm gym-workout. The main idea of our approach is to merge joint kinematic data in-between sEMG channels and visualize as one image which will serve as an input to the CNN. The following research questions represent the motivation for this study:

- "If muscle physiologic data is visualized together with kinematic data as a spatio-temporal image, can a relationship between these two data be imputed by the CNN since it consists of an automatic feature extractor followed by a classifier?"
- "If the input image is manipulated to show more distinctive spatio-temporal features from a point of view of human eyes, then how this manipulation does affect to the CNN performances in terms of training loss and test accuracy?"

The above motivation leads us to the following validation procedure. First, the measured sEMG and joint angle data set is augmented using amplitude adjusted Fourier transform to generate the surrogate data while consistency is being sustained. Next, we visualize the measured sEMG and joint angle data as one heatmap by setting a horizontal and a vertical axis to be the time and the channels of the measured data, respectively. Then, we utilize the heatmap as an input data to CNN and then investigate the classification accuracy via CNN varies according to the merging location of the joint angle, e.g., in-between sEMG channels (just as a separate bar) or at the bottom of all sEMG channels. Furthermore, the measured sEMG and joint angle are also visualized with the patches of a Hadamard product matrix made of the joint angle and each sEMG channel, which is believed that, at the first glance, spatio-temporal features become challenging to be recognized. Finally, we perform statistical analysis to substantiate the main effects of the human-eye friendly image manipulation and the level of data augmentation as well as the interaction effect between those two.

To our best knowledge, the idea of the merging of muscle physiology and joint kinematics as well as the human-eye friendly image manipulation have not yet been fully studied in a field of CNN-based classification. Therefore, the contribution of this study can be summarized as follows:

- We propose a novel approach for upper arm gym-workout classification by imputing a relationship between muscle physiology and joint kinematics via CNN feature extraction.
- We introduce a data augmentation technique for time series, present various visualization methods according to human-eye friendly image manipulation, and statistically analyze the CNN classification performance based on experimental evaluations.
- The outcomes from this study can be utilized to advance the state of the art in the problem of developing exercise monitoring systems by providing the level of data augmentation and the visualization method which guarantees the best CNN classification performance.

The rest of the paper is organized as follows: the proposed approach for gym-workout classification is introduced in Section 2; specifically, our approach is explained through corresponding subsections as follows: system architecture, experimental setup and procedure, data augmentation technique, post data-processing, and is culminated to the main idea about imputing a biopotential-kinematic relationship and performing classification via CNN. In Section 3, the classification result is reported in terms of training loss and test accuracy, which is followed by statistical analysis. In Section 4, significant outcomes and findings are discussed. Lastly, Section 5 will be the conclusion of this paper.

## 2. Methods

### 2.1. System Architecture, Experimental Setup and Protocol

Figure 1 depicts the overall system architecture, data measurement and flows, data processing, and CNN training for the proposed upper arm gym-workout classification.

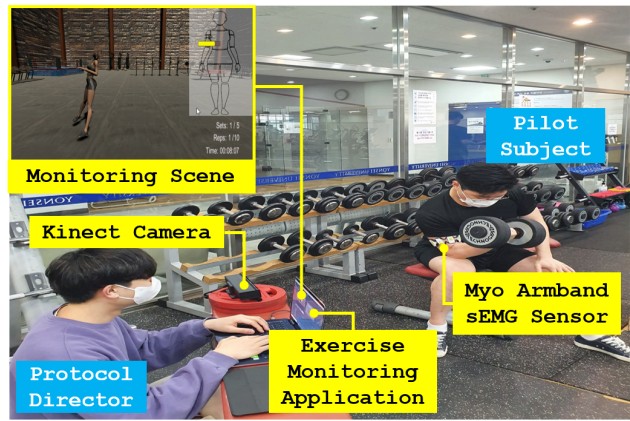

(**a**) Pilot subject demonstration

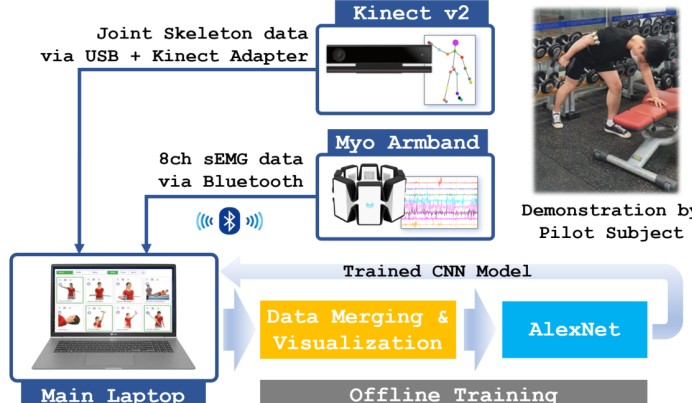

(**b**) Data measurement by Kinect v2 and Myo armband.

**Figure 1.** The overall system architecture: the measured data from Kinect v2 and Myo armband are merged and visualized. The visualized image is used as an input for CNN training. Both a pilot subject and a protocol director are wearing a mask by following the government quarantine instruction against COVID-19.

Our exercise monitoring system consisted of a laptop with a custom-developed Unity3D-based software application, an armband type 8-channel sEMG sensor (Myo armband, Thalmic Labs, Brooklyn, NY, USA), a RGB-d camera (Kinect v2, Microsoft, Redmond, WA, USA). Throughout the paper, the sEMG sensor and the RGB-d camera will be referred to as more common name: Myo armband and Kinect v2.

A professional trainer was recruited as a pilot subject (PS). The Myo armband was mounted on PS's right upper arm under a protocol director (PD)'s supervision. Figure 2 shows two exercises demonstrated by the PS and the 8-channel deployment on his right upper arm. After being equipped with Myo armband, the PS was guided by the PD in front of Kinect v2 and given instructions about experimental procedure and safety. The PS was instructed to perform a dumbbell curl (target muscle: biceps brachii) at the first visit and a dumbbell kickback (target muscle: triceps brachii) at the second visit. The experimental protocol was as follows:

1.  According to PD's hand signal at every 1.6 s, the PS performed a dumbbell curl one time (one trial).
2.  After performing 10 trials, the PS was supposed to take a 15 min break (the end of one session) to minimize the effect of muscle fatigue.
3.  Repeat six sessions.

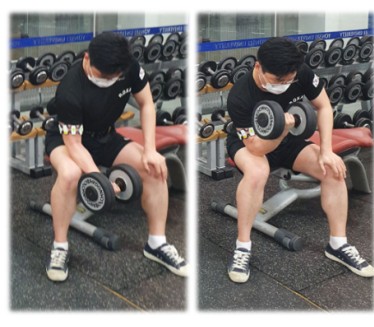
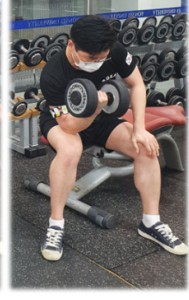

(**a**) Dumbbell curl

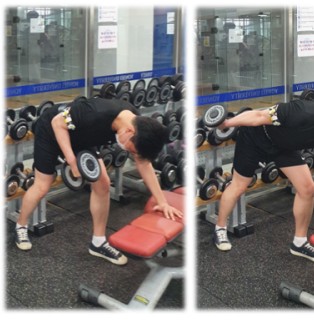
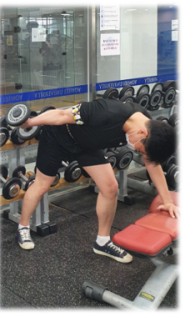

(**b**) Dumbbell kickback

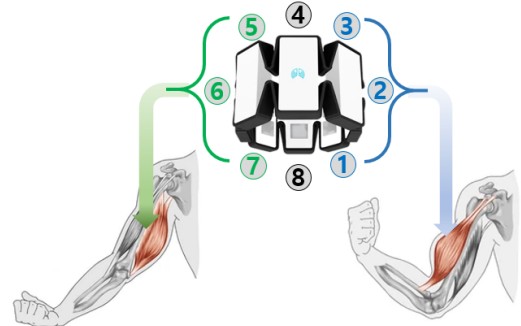

(**c**) Myo armband 8-channel deployment

**Figure 2.** The pilot subject demonstrates dumbbell curl and dumbbell kickback. For Myo armband Ch.1∼Ch.3 and Ch.5∼Ch.7 correspond to biceps and triceps brachii, respectively. Ch.4 and Ch.8 are placed on boundaries between two muscles.

While the PS was performing the exercise, sEMG data and joint skeleton data were sent via Bluetooth and USB+Kinect adapter connections, respectively. The sampling rate of Myo armband was set to 20 ms; For each channel, hence, 80 time series data samples

were recorded for one trial (corresponds to 1.6 s). An elbow joint angle was calculated by VITRUVIUS using skeleton data from Kinect v2 [10]. The same sampling rate was applied to Kinect v2; therefore, 80 data samples were stored for the elbow joint angle. After two days from the first visit, dumbbell kickback was measured under the same experimental protocol as well.

### 2.2. Data Augmentation Using AAFT and Signal Processing

In contrast to mechanical or electrical signal measurement, it is difficult to measure a large number of consistent sEMG signals for a repetitive task due to muscle fatigue. To prevent either under- or over-fitting and increase the generalization ability of a neural network, therefore, a proper data augmentation technique must be applied. The measured both sEMG and joint angle data are augmented by adopting the amplitude adjusted Fourier transform (AAFT) (Python code for AAFT can be found at https://github.com/manu-mannattil/nolitsa, accessed on 2 February 2021 [50]). We here recapitulate AAFT by the following [51]:

1. Generate a Gaussian sequence, say $y(n)$, using pseudo-random generator;
2. Reorder $y(n)$ according to the rank of the measured original data $x(n)$, say this reordered sequence $z(n)$;
3. Perform Fourier transform to $z(n)$:

$$Y(k) = \sum_{n=0}^{N-1} z(n)e^{-j2\pi nk/N}$$

4. Randomize phase: $Y'(k) = Y(k)e^{j\phi}$, where

When data are even: $\begin{cases} \phi(f_0) = 0 \\ \phi(f_i) = -\phi(f_k), \quad i = 2 - \frac{N}{2}, \, k = N - \frac{N}{2} + 1 \\ \phi(f_{N/2}) = 0 \end{cases}$

When data are odd: $\begin{cases} \phi(f_0) = 0 \\ \phi(f_i) = -\phi(f_k), \quad i = 2 - \frac{N+1}{2}, \, k = N - \frac{N+1}{2} + 1 \end{cases}$

5. Perform inverse Fourier transform:

$$y'(n) = \frac{1}{N} \sum_{k=1}^{N-1} Y'(k)e^{j2\pi nk/N}$$

6. According to the rank of $y'(n)$, reorder the measured original data $x(n)$; this yields the surrogate data $x'(n)$.

The AAFT generates the surrogates of the original data in terms of temporal correlation, amplitude distribution, power spectral density [51,52]. For the sEMG measurement, since it is rather difficult to obtain a large amount of consistent data due to muscle fatigue, the AAFT is frequently used for data augmentation. In this study, the AAFT was employed with the same purpose and similar application can be found in [53]. We note that a ratio between the number of data after being augmented and that of the original data will be referred to as the level of data augmentation. For instance, if the total number of data was tripled, then the level of data augmentation was 3 and will be denoted by AAFT(3).

After data augmentation by AAFT, moving root mean square (RMS) filtering was applied to both sEMG and joint angle data (original and surrogates) with a sliding window length 10. Recall that each sEMG channel as well as the elbow joint angle data contained 80 time series data samples. These data were integrated over every five samples, which yielded $8 \times 16$ for the 8-channel sEMG data and $1 \times 16$ for the elbow joint angle data, respectively. The $8 \times 16$ sEMG data were intrachannel normalized and the $1 \times 16$ elbow joint angle data were interchannel normalized.

### 2.3. Imputing a Muscle Activation to Joint Kinematics Relationship and Classification via CNN

Figure 3 illustrates the examples of $\mathcal{I}_{\text{easy}}$, $\mathcal{I}_{\text{fair}}$, and $\mathcal{I}_{\text{chal}}$, which were produced by our proposed approach introduced below. The subscript of $\mathcal{I}$—easy, fair, and chal (=challenging)—represents the level of manipulation for the recognizability of input image features from human eyes' point of view. For example, from Figure 3a, we could easily distinguish dumbbell curl and dumbbell kickback by the spectral color of upper- or lower-stripes. In Figure 3b, in contrast, the bottom stripe seemed to be related to the upper stripes but separated; especially, in case of dumbbell curl. Indeed, Figure 3a was produced by interleaving the interchannel normalized elbow joint angle data in-between biceps and triceps channels, whereas the elbow joint angle data were simply added at the bottom of the 8-channel sEMG data. From Figure 3c, we can see diagonal patterns but the patterns themselves are rather challenging to be described.

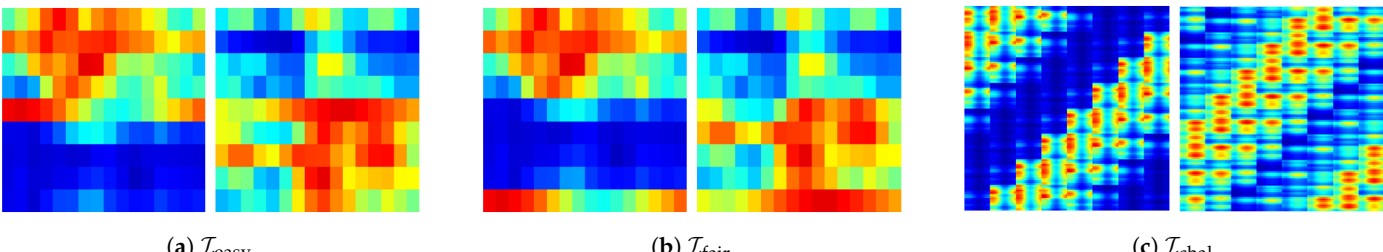

(**a**) $\mathcal{I}_{\text{easy}}$　　　　　　　　　　　(**b**) $\mathcal{I}_{\text{fair}}$　　　　　　　　　　　(**c**) $\mathcal{I}_{\text{chal}}$

**Figure 3.** The examples of $\mathcal{I}_{\text{easy}}$, $\mathcal{I}_{\text{fair}}$, and $\mathcal{I}_{\text{chal}}$: For all (**a**–**c**), a pair of figures represents dumbbell curl (**left**) versus dumbbell kickback (**right**).

The underlying rationale of the above manipulation was that we wanted to know whether producing an input image data set with considering human eyes' point of view had an effect on classification via CNN or not. Especially, we want to investigate how it affected feature extraction for imputing a muscle activation to joint kinematics relationship, which could be implied by the training loss and the test accuracy. In this sense, we summarized the feature characteristics of $\mathcal{I}_{\text{easy}}$, $\mathcal{I}_{\text{fair}}$, $\mathcal{I}_{\text{chal}}$ as follows:

- $\mathcal{I}_{\text{easy}}$: an image contained features that were easy to be recognized by human eyes, e.g., geometry, color, etc.
- $\mathcal{I}_{\text{fair}}$: an image contained features that were fair to be recognized by human eyes, e.g., simple rules, simple pattern, local differences, etc.
- $\mathcal{I}_{\text{chal}}$: an image contained features that were challenging to be recognized by human eyes, e.g., mathematically defined patterns such as correlation, attraction, bifurcation, fractal, etc.

$\mathcal{I}_{\text{easy}}$, $\mathcal{I}_{\text{fair}}$, $\mathcal{I}_{\text{chal}}$ can be produced according to manipulations introduced below. Let $s_m(n)$ and $a(n)$ be the $m$th channel sEMG data and the elbow joint angle data after signal processing, respectively. Recall that both data were of $\mathbb{R}^{1 \times 16}$. First, $\mathcal{I}_{\text{easy}}$ was defined to be

$$\mathcal{I}_{\text{easy}}(m,n) = \begin{bmatrix} s_1(n) \\ \vdots \\ s_4(n) \\ a(n) \\ s_5(n) \\ \vdots \\ s_8(n) \end{bmatrix} = \begin{bmatrix} s_1(1) & s_1(2) & \cdots & s_1(16) \\ \vdots & \vdots & \ddots & \vdots \\ s_4(1) & s_4(2) & \cdots & s_4(16) \\ a(1) & a(2) & \cdots & a(16) \\ s_5(1) & s_5(2) & \cdots & s_5(16) \\ \vdots & \vdots & \ddots & \vdots \\ s_8(1) & s_8(2) & \cdots & s_8(16) \end{bmatrix} \in \mathbb{R}^{9 \times 16}. \tag{1}$$

Namely, $a(n)$ was interleaved in-between the biceps and triceps channels of the sEMG. Second, we defined $\mathcal{I}_{\text{fair}}$ as

$$\mathcal{I}_{\text{fair}}(m,n) = \begin{bmatrix} s_1(n) \\ s_2(n) \\ \vdots \\ s_8(n) \\ a(n) \end{bmatrix} = \begin{bmatrix} s_1(1) & s_1(2) & \cdots & s_1(16) \\ s_2(1) & s_2(2) & \cdots & s_2(16) \\ \vdots & \vdots & \ddots & \vdots \\ s_8(1) & s_8(2) & \cdots & s_8(16) \\ a(1) & a(2) & \cdots & a(16) \end{bmatrix} \in \mathbb{R}^{9 \times 16}. \tag{2}$$

Thus, $a(n)$ was merged at the bottom of the sEMG channels. Third, to define $\mathcal{I}_{\text{chal}}$, we first defined matrices $S_m$ and $A$ which could be generated by replicating $s_m(n)$ and $a(n)$ to row- and column-wise, respectively:

$$S_m = \begin{bmatrix} s_m(1) & s_m(2) & \cdots & s_m(16) \\ s_m(1) & s_m(2) & \cdots & s_m(16) \\ \vdots & \vdots & \ddots & \vdots \\ s_m(1) & s_m(2) & \cdots & s_m(16) \end{bmatrix} \quad \text{and} \quad A = \begin{bmatrix} a(1) & a(1) & \cdots & a(1) \\ a(2) & a(2) & \cdots & a(2) \\ \vdots & \vdots & \ddots & \vdots \\ a(16) & a(16) & \cdots & a(16) \end{bmatrix}. \tag{3}$$

Both $S_m$ and $A$ are of $\mathbb{R}^{16 \times 16}$. Now, we can define $\mathcal{I}_{\text{chal}}$ by

$$\mathcal{I}_{\text{chal}} = \begin{bmatrix} S_1 \odot A & S_2 \odot A & \cdots & S_8 \odot A \\ S_2 \odot A & S_3 \odot A & \cdots & S_1 \odot A \\ \vdots & \vdots & \ddots & \vdots \\ S_8 \odot A & S_1 \odot A & \cdots & S_7 \odot A \end{bmatrix} \in \mathbb{R}^{128 \times 128} \tag{4}$$

where $\odot$ represents the Hadamard product. Namely, $\mathcal{I}_{\text{chal}}$ consists of patches of $S_m \odot A$. Finally, all $\mathcal{I}_{\text{easy}}$, $\mathcal{I}_{\text{fair}}$, and $\mathcal{I}_{\text{chal}}$ were visualized as $227 \times 227 \times 3$ (width×height×RGB-channels) heatmaps by upscaling from $9 \times 16 \times 3$, $9 \times 16 \times 3$, and $128 \times 128 \times 3$, respectively. For this upscaling, bilinear interpolation was used. The selected examples of $\mathcal{I}_{\text{easy}}$, $\mathcal{I}_{\text{fair}}$, $\mathcal{I}_{\text{chal}}$ are presented in Figure 4.

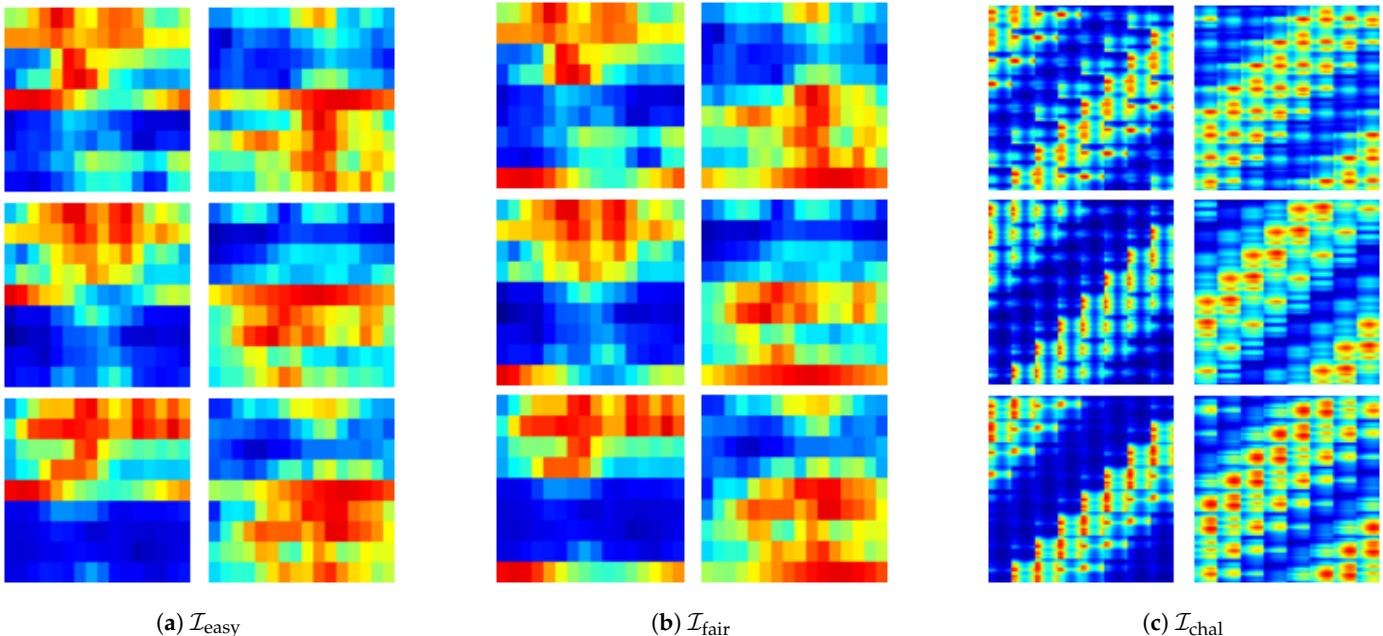

(a) $\mathcal{I}_{\text{easy}}$        (b) $\mathcal{I}_{\text{fair}}$        (c) $\mathcal{I}_{\text{chal}}$

**Figure 4.** The selected examples of produced $\mathcal{I}_{\text{easy}}$, $\mathcal{I}_{\text{fair}}$, and $\mathcal{I}_{\text{chal}}$: For all (**a**–**c**), a pair of figures represents dumbbell curl (**left**) versus dumbbell kickback (**right**). For (**a**), it is easy to see that the left and the right figures can be featured based on upper and lower stripes of the heat map. From (**b**), stripe pattern in the left figure is separated at top and bottom; nevertheless, the two figures can be fairly featured with stripe patterns. In (**c**), the two figures show the opposite diagonal patterns against each other. The pattern seems like periodic but becomes challenging to be featured in a simple way.

To train a model and classify the visualized dumbbell curl and dumbbell kickback, the AlexNet was employed [54]. Figure 5 shows gym-workout classification via AlexNet with an example of $\mathcal{I}_{\text{easy}}$ as an input image. According to a combination of the 5-level of data augmentation by AAFT and the 3-level of manipulation for the recognizability of input image features, there existed 15 conditions. Under each condition, training and classification via CNN (AlexNet) were repeated 10 times with shuffling the sequence of input images for each repetition. The performance of CNN is recorded in terms of the training loss and the test accuracy at the end of each repetition.

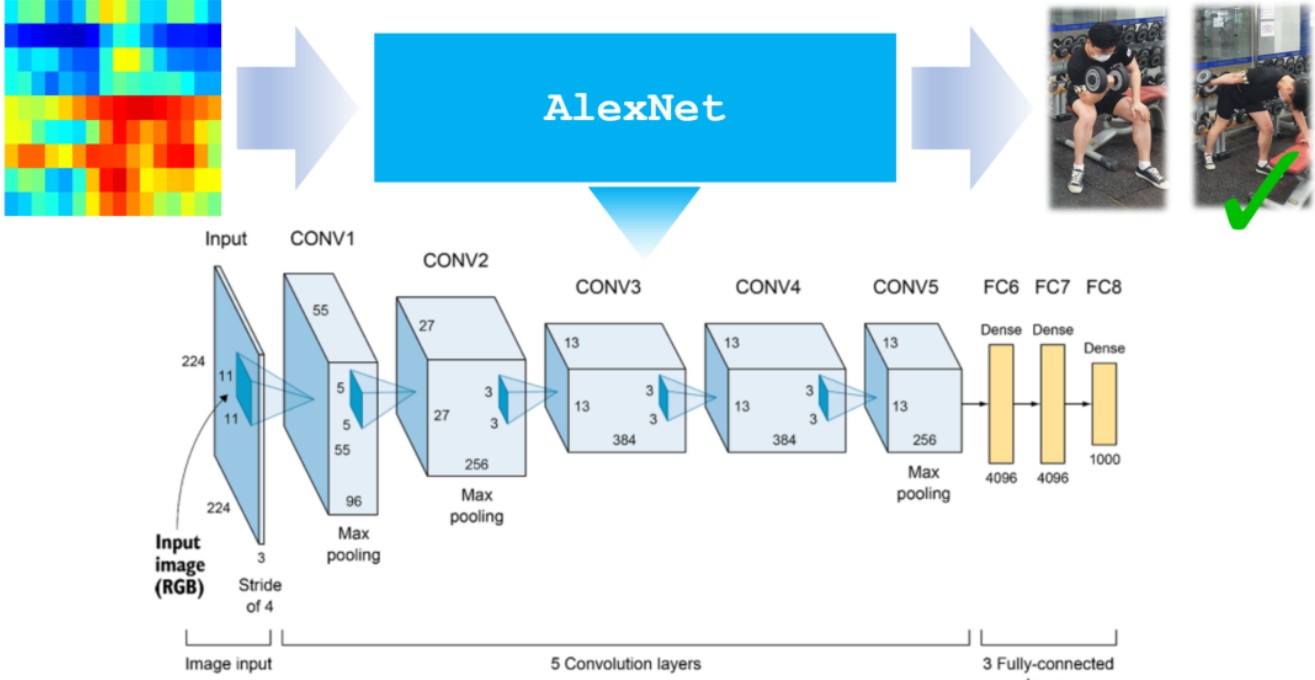

**Figure 5.** An example of gym-workout classification via AlexNet with an input image $\mathcal{I}_{\text{easy}}$ [54].

### 2.4. Statistical Analysis and Research Questions

For statistical analysis, the level of data augmentation by AAFT was set to one dependent factor and represented by AAFT(1), AAFT(2), AAFT(3), AAFT(4), AAFT(5). The representation AAFT(4) represents that the original data were augmented by quadruple. The other dependent factor was set to the level of manipulation for the recognizability of input image features which were represented by $\mathcal{I}_{\text{easy}}$, $\mathcal{I}_{\text{fair}}$, $\mathcal{I}_{\text{chal}}$. Statistical analysis was performed to explain the effects of the two factors (either positive or negative) on CNN (AlexNet) performance in terms of the training loss and the test accuracy.

A repeated measures analysis of variance (rANOVA) was performed to identify the effect of two dependent factors with the significance level of $p < 0.05$ (IBM SPSS Statistics, v25, Chicago, IL, USA). If the assumption of sphericity was violated for the main effects, the degree of freedom was corrected using Greenhouse-Geisser estimates of sphericity. The Bonferroni-adjusted pairwise comparison was used, and the result was reported in the form of "mean difference (standard error)." Meanwhile, statistically analyzing the CNN performance, the following research questions were addressed:

**(Q1)** "How did the level of data augmentation by AAFT affect CNN performance?"
**(Q2)** "How did the level of manipulation for the recognizability of input image features affect CNN performance?"
**(Q3)** "What was the optimal combination of the two factors for the best CNN performance?"

(Q1) and (Q2) could be answered by investigating the main effect of the level of data augmentation by AAFT and the level of manipulation for the recognizability of input image

features, respectively. The answer for (Q3) could be found by observing the interaction effect between the two factors as well as pairwise comparison result with corresponding mean and standard deviation values.

## 3. Results

### 3.1. Overall CNN Performance

To evaluate CNN performance, training:validation:test sets were organized by 8:1:1 ratio. For instance, under AAFT(1) which corresponded to no data augmentation, $\mathcal{I}_{easy}$, $\mathcal{I}_{fair}$, $\mathcal{I}_{chal}$ included 60 images each; accordingly, the number of data in training:validation:test sets were 48:6:6. For each condition of AAFT(*) and $\mathcal{I}_*$, the evaluation was repeated 10 times. The data set was shuffled for every repetition. Tables 1 and 2 show that the mean and standard deviation of training loss and test accuracy according to two dependent factors, respectively. Mean values presented in bold face represent the best performance (the lowest value for training loss and the highest value for test accuracy).

For better readability, we summarize the estimated mean values and pairwise comparison results of the two CNN performance metrics across the five levels of data augmentation by AAFT and the three levels of manipulation for the recognizability of input image features in Table 3. The explanation about the statistical analysis result of Table 3 is followed in Sections 3.2–3.4.

**Table 1.** The mean and standard deviation of training loss.

| Training Loss | AAFT(1) † | | | AAFT(2) | | | AAFT(3) | | | AAFT(4) | | | AAFT(5) | | |
|---|---|---|---|---|---|---|---|---|---|---|---|---|---|---|---|
| | $\mathcal{I}_{easy}$ | $\mathcal{I}_{fair}$ | $\mathcal{I}_{chal}$ | $\mathcal{I}_{easy}$ | $\mathcal{I}_{fair}$ | $\mathcal{I}_{chal}$ | $\mathcal{I}_{easy}$ | $\mathcal{I}_{fair}$ | $\mathcal{I}_{chal}$ | $\mathcal{I}_{easy}$ | $\mathcal{I}_{fair}$ | $\mathcal{I}_{chal}$ | $\mathcal{I}_{easy}$ | $\mathcal{I}_{fair}$ | $\mathcal{I}_{chal}$ |
| 1st run | 3.38 | 2.93 | 3.36 | 2.96 | 2.62 | 3.37 | 2.63 | 2.62 | 3.37 | 1.62 | 2.27 | 1.57 | 1.11 | 1.66 | 1.58 |
| 2nd run | 3.19 | 3.36 | 3.27 | 3.20 | 3.12 | 3.48 | 2.01 | 1.86 | 3.74 | 1.86 | 2.54 | 2.31 | 1.46 | 1.11 | 2.21 |
| 3rd run | 3.07 | 3.21 | 3.39 | 3.04 | 3.56 | 3.01 | 3.30 | 3.01 | 2.62 | 1.83 | 1.54 | 2.60 | 1.61 | 1.10 | 1.63 |
| 4th run | 3.43 | 3.17 | 3.42 | 3.55 | 3.66 | 3.39 | 2.58 | 2.12 | 3.14 | 2.66 | 2.35 | 2.17 | 1.01 | 1.03 | 1.04 |
| 5th run | 3.46 | 3.38 | 2.79 | 2.85 | 2.45 | 3.41 | 2.09 | 2.11 | 3.25 | 2.52 | 2.04 | 2.26 | 1.14 | 0.82 | 1.27 |
| 6th run | 3.38 | 3.20 | 3.25 | 3.29 | 3.12 | 3.51 | 2.73 | 2.42 | 2.91 | 2.58 | 1.30 | 2.50 | 1.79 | 1.32 | 1.18 |
| 7th run | 3.46 | 3.36 | 2.44 | 2.73 | 3.06 | 3.63 | 2.96 | 1.78 | 3.03 | 1.96 | 1.23 | 3.36 | 0.96 | 0.32 | 1.21 |
| 8th run | 3.80 | 3.46 | 2.75 | 3.47 | 3.20 | 3.60 | 2.12 | 2.52 | 2.36 | 1.28 | 1.23 | 2.56 | 1.53 | 1.24 | 2.01 |
| 9th run | 3.25 | 3.30 | 3.14 | 2.83 | 2.72 | 3.38 | 2.20 | 0.44 | 2.68 | 2.18 | 1.22 | 3.07 | 1.28 | 1.85 | 2.23 |
| 10th run | 3.29 | 3.46 | 3.23 | 2.75 | 2.64 | 2.52 | 2.90 | 2.80 | 2.37 | 2.61 | 2.45 | 1.81 | 1.48 | 1.39 | 1.38 |
| **Mean** | 3.37 | **3.28** | 3.10 | **3.07** | 3.13 | 3.27 | 2.55 | **2.17** | 2.95 | 2.11 | **1.82** | 2.42 | 1.34 | **1.18** | 1.57 |
| **Std** | 0.20 | **0.16** | 0.33 | **0.30** | 0.44 | 0.37 | 0.44 | **0.73** | 0.45 | 0.48 | **0.56** | 0.53 | 0.28 | **0.43** | 0.44 |

†: AAFT(1) corresponds to no data augmentation.

**Table 2.** The mean and standard deviation of test accuracy (%).

| Test Accuracy | AAFT(1) † | | | AAFT(2) | | | AAFT(3) | | | AAFT(4) | | | AAFT(5) | | |
|---|---|---|---|---|---|---|---|---|---|---|---|---|---|---|---|
| | $\mathcal{I}_{easy}$ | $\mathcal{I}_{fair}$ | $\mathcal{I}_{chal}$ | $\mathcal{I}_{easy}$ | $\mathcal{I}_{fair}$ | $\mathcal{I}_{chal}$ | $\mathcal{I}_{easy}$ | $\mathcal{I}_{fair}$ | $\mathcal{I}_{chal}$ | $\mathcal{I}_{easy}$ | $\mathcal{I}_{fair}$ | $\mathcal{I}_{chal}$ | $\mathcal{I}_{easy}$ | $\mathcal{I}_{fair}$ | $\mathcal{I}_{chal}$ |
| 1st run | 100.0 | 100.0 | 50.0 | 91.7 | 50.0 | 50.0 | 97.2 | 94.4 | 50.0 | 100.0 | 100.0 | 99.0 | 100.0 | 100.0 | 85.0 |
| 2nd run | 100.0 | 50.0 | 70.8 | 100.0 | 100.0 | 50.0 | 91.7 | 100.0 | 98.6 | 100.0 | 100.0 | 50.0 | 100.0 | 100.0 | 99.2 |
| 3rd run | 66.7 | 95.8 | 50.0 | 100.0 | 50.0 | 50.0 | 100.0 | 98.6 | 50.0 | 97.9 | 99.0 | 62.5 | 100.0 | 100.0 | 100.0 |
| 4th run | 100.0 | 50.0 | 50.0 | 100.0 | 50.0 | 50.0 | 100.0 | 100.0 | 50.0 | 100.0 | 100.0 | 69.8 | 100.0 | 100.0 | 100.0 |
| 5th run | 100.0 | 87.5 | 50.0 | 100.0 | 100.0 | 50.0 | 100.0 | 100.0 | 96.9 | 100.0 | 100.0 | 96.9 | 100.0 | 100.0 | 100.0 |
| 6th run | 88.9 | 100.0 | 50.0 | 89.6 | 97.9 | 75.0 | 100.0 | 100.0 | 50.0 | 100.0 | 100.0 | 99.0 | 100.0 | 100.0 | 99.2 |
| 7th run | 66.7 | 100.0 | 50.0 | 100.0 | 89.6 | 97.9 | 98.6 | 100.0 | 50.0 | 99.0 | 100.0 | 99.0 | 100.0 | 100.0 | 100.0 |
| 8th run | 97.2 | 100.0 | 50.0 | 89.6 | 100.0 | 70.8 | 98.6 | 100.0 | 91.7 | 100.0 | 100.0 | 50.0 | 96.7 | 100.0 | 98.3 |
| 9th run | 38.9 | 50.0 | 50.0 | 93.8 | 54.2 | 50.0 | 100.0 | 100.0 | 54.2 | 100.0 | 100.0 | 54.2 | 100.0 | 99.2 | 82.5 |
| 10th run | 86.1 | 100.0 | 50.0 | 97.9 | 83.3 | 91.7 | 98.6 | 100.0 | 94.4 | 99.0 | 97.9 | 89.6 | 100.0 | 100.0 | 94.2 |
| **Mean** | **84.4** | 83.3 | 52.1 | **96.3** | 77.5 | 63.5 | 98.5 | **99.3** | 68.9 | 99.6 | **99.7** | 76.6 | 99.7 | **99.9** | 95.8 |
| **Std** | **20.7** | 23.3 | 6.6 | **4.6** | 23.4 | 19.0 | 2.6 | **1.8** | 23.6 | 0.7 | **0.7** | 22.2 | 1.1 | **0.3** | 6.6 |

†: AAFT(1) corresponds to no data augmentation.

**Table 3.** Two metrics of convolutional neural network (CNN) performance. Value represents the mean for the five levels of data augmentation by AAFT (Factor 1) and the three levels of manipulation for the recognizability of input image features (Factor 2), and *p*-values for interaction effect. Superscripts represent significant differences from the other main effects conditions resulting from Bonferroni adjusted pairwise comparison.

| Performance | Factor 1 | | | | | Factor 2 | | | Interaction |
|---|---|---|---|---|---|---|---|---|---|
| Metrics for CNN | AFFT(1) | AFFT(2) | AFFT(3) | AFFT(4) | AFFT(5) | $\mathcal{I}_\text{easy}$ | $\mathcal{I}_\text{fair}$ | $\mathcal{I}_\text{chal}$ | *p*-Value |
| Training Loss | 3.25 [(3),(4),(5)] | 3.16 [(3),(4),(5)] | 2.56 [(1),(2),(5)] | 2.12 [(1),(2),(5)] | 1.37 [(1),(2),(3),(4)] | 2.49 | 2.32 | 2.66 | **0.025** |
| Test Accuracy (%) | 73.3 [(3),(4),(5)] | 79.1 [(5)] | 88.9 [(1),(5)] | 91.9 [(1)] | 98.5 [(1),(2),(3)] | 95.7 [c] | 91.9 [c] | 71.4 [e,f] | **0.009** |

### 3.2. Effects of the Level of Data Augmentation by AAFT

Effect on training loss: the rANOVA result indicated that there was a significant main effect of the levels of data augmentation by AAFT on training loss, $F(4, 18.26) = 85.93$, $p < 0.001$. The pairwise comparison revealed that AAFT(1) vs. AAFT(2) yields mean(standard error) = 0.10(0.08), $p = 1.000$, AAFT(1) vs. AAFT(3), 0.70(0.11), $p = 0.001$, AAFT(1) vs. AAFT(4), 1.14(0.08), $p < 0.001$, and AAFT(1) vs. AAFT(5), 1.89(0.09), $p < 0.001$, indicating a significant difference between AAFT(1) and the other levels of data augmentaion except AAFT(2). This means that the data augmentation had an effect on the training loss when greater or equal than triple. In addition, the comparisons of AFFT(2) vs. other greater levels showed significant differences as AFFT(2) vs. AFFT(3), 0.60(0.13), $p = 0.010$, AFFT(2) vs. AFFT(4), 1.04(1.22), $p < 0.001$, and AFFT(2) vs. AFFT(5), 1.79(1.30), $p < 0.001$, implying that the training loss significantly decreased if the level of data augmentation was greater or equal than triple. Between AFFT(3) and AFFT(4), no significant differences were found. AFFT(5) showed significance difference against all the other levels.

Effect on test accuracy: Please note that test accuracy was presented in terms of percentile in Table 2; therefore, 100% corresponds to 1.00 from now on. Analysis on test accuracy yielded a significant main effect for the level of data augmentation by AAFT on test accuracy as well, $F(4, 0.31) = 18.50$, $p < 0.001$. Pairwise comparisons revealed that AAFT(1) vs. AAFT(2), $-0.06(0.04)$, $p = 1.000$, AAFT(1) vs. AAFT(3), $-0.16(0.40)$, $p = 0.017$, AAFT(1) vs. AAFT(4), $-0.19(0.03)$, $p = 0.002$, and AAFT(1) vs. AAFT(5), $-0.25(0.03)$, $p < 0.001$, suggesting that the test accuracy increased significantly when the level of data augmentation was greater or equal than triple compared to AAFT(1). There were no significant differences among the test accuracy under AAFT(2), AAFT(3), AAFT(4). In contrast, AAFT(5) indicates significant difference over AAFT(1), 0.25(0.03), $p < 0.001$, AAFT(2), 0.19(0.04), $p = 0.004$, AAFT(3) 0.10(0.02), $p = 0.032$, except AAFT(4). This implies the test accuracy improved significantly as data were augmented by AAFT.

### 3.3. Effects of the Level of Manipulation for the Recognizability of Input Image Features

Effect on training loss: The level of manipulation for the recognizability of input image features showed the significant main effect on training loss, $F(2, 1.50) = 5.94$, $p = 0.010$. However, pairwise comparison revealed no significant difference among $\mathcal{I}_\text{easy}$, $\mathcal{I}_\text{fair}$, and $\mathcal{I}_\text{chal}$. The comparison of $\mathcal{I}_\text{fair}$ vs. $\mathcal{I}_\text{chal}$ only showed marginal tendency, $-0.35(0.12)$, $p = 0.062$.

Effect on test accuracy: The significant main effect of the level of manipulation for the recognizability of input image features on test accuracy was also found, $F(2, 0.86) = 72.21$, $p < 0.001$. Pairwise comparison yielded a significant difference for both $\mathcal{I}_\text{easy}$ vs. $\mathcal{I}_\text{chal}$, 0.04(0.02), $p < 0.001$, and $\mathcal{I}_\text{fair}$ vs. $\mathcal{I}_\text{chal}$, 0.21(0.02), $p < 0.001$, which indicated that both $\mathcal{I}_\text{easy}$ and $\mathcal{I}_\text{fair}$ yielded better test accuracy over $\mathcal{I}_\text{chal}$. No significant difference, however, was found between $\mathcal{I}_\text{easy}$ and $\mathcal{I}_\text{fair}$.

### 3.4. Interaction Effects of the Level of Data Augmentation by AAFT × the Level of Manipulation for the Recognizability of Input Image Features

Interaction effect on training loss: The analysis revealed that there was a significant interaction effect between the level of data augmentation by AAFT and the level of manipulation for the recognizability of input image features in training loss, $F(8, 0.40) = 2.43$, $p = 0.022$, as shown in the rightmost column in Table 3. From Figure 6a, we can see that the train losses of both $\mathcal{I}_{\text{easy}}$ and $\mathcal{I}_{\text{fair}}$ were greater than that of $\mathcal{I}_{\text{chal}}$ under AAFT(1). They tended to be decreased quickly and became less than that of $\mathcal{I}_{\text{chal}}$ as the level of data augmentation increased. This indicates that the level of manipulation for the recognizability of input image features might have had a different effect on training loss depending on $\mathcal{I}_{\text{easy}}$, $\mathcal{I}_{\text{fair}}$, and $\mathcal{I}_{\text{chal}}$. Pairwise comparison revealed a marginal tendency for $\mathcal{I}_{\text{fair}}$ vs. $\mathcal{I}_{\text{chal}}$ under AAFT(5), $-0.39(0.14)$, $p = 0.061$, but no significant difference was found.

Interaction effect on test accuracy: A significant interaction effect between the level of data augmentation by AAFT and the level of manipulation for the recognizability of input image features was also found in test accuracy, $F(8, 0.06) = 2.80$, $p = 0.009$, as reported in the rightmost column in Table 3. Figure 6b depicts that the test accuracy mostly tended to be better as AAFT level increased. In addition, the test accuracy of both $\mathcal{I}_{\text{easy}}$ and $\mathcal{I}_{\text{fair}}$ tended to be more precise than $\mathcal{I}_{\text{chal}}$ across all AAFT levels. Pairwise comparison indeed revealed that the test accuracy of both $\mathcal{I}_{\text{easy}}$ and $\mathcal{I}_{\text{fair}}$ was significantly higher than that of $\mathcal{I}_{\text{chal}}$ upto AAFT(4). No significant difference was found between $\mathcal{I}_{\text{easy}}$ and $\mathcal{I}_{\text{fair}}$ across all AAFT levels.

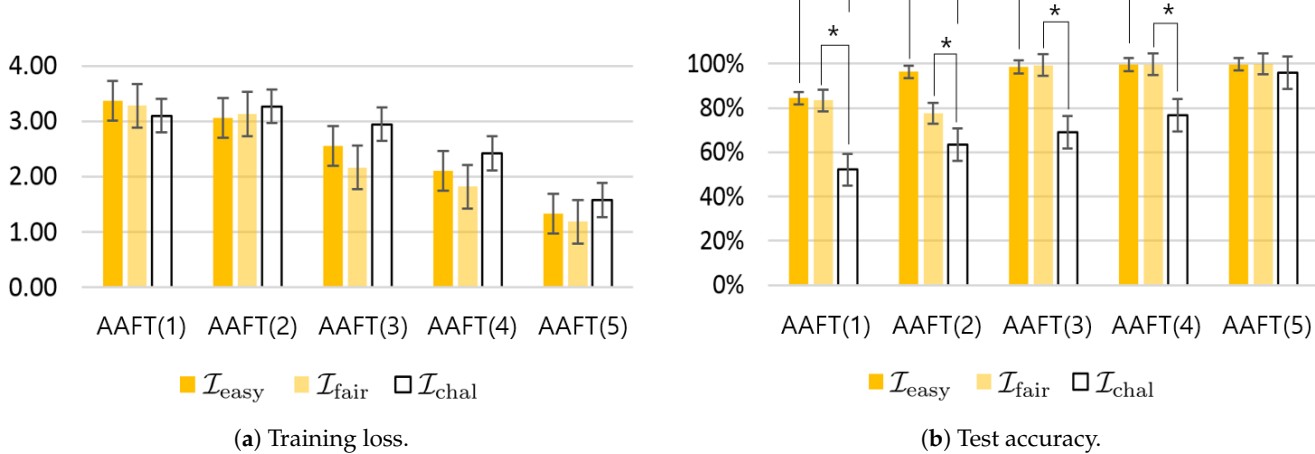

(**a**) Training loss.                                   (**b**) Test accuracy.

**Figure 6.** Mean values for (**a**) training loss and (**b**) test accuracy: For (**a**), The training loss shows several tendencies with respect to two factors, but no significant difference found by pairwise comparison. For (**b**), pairwise comparison indeed revealed that the test accuracy of both $\mathcal{I}_{\text{easy}}$ and $\mathcal{I}_{\text{fair}}$ was significantly higher than that of $\mathcal{I}_{\text{chal}}$ upto AAFT(4). No significant difference was found between $\mathcal{I}_{\text{easy}}$ and $\mathcal{I}_{\text{fair}}$ across all AAFT levels. Significance is marked for $p < 0.05$ ($*$).

### 3.5. Comparison Result for Various Neural Network Models

The comparison results for AlexNet and three layer (input-hidden-output) neural networks (NNs) are presented in Tables 4 and 5. h1 through h4 represent the number of hidden nodes in the hidden layer. To evaluate NNs, $\mathcal{I}_{\text{easy}}$ and $\mathcal{I}_{\text{fair}}$ (which consist of $9 \times 16$ real values) were flattened to $144 \times 1$ vectors, then applied to the NN as an input. Similarly, $128 \times 128$ $\mathcal{I}_{\text{chal}}$ was converted into $16,384 \times 1$ vector. Accordingly, the number of input nodes was set to 144, 144, and 16,384 for $\mathcal{I}_{\text{easy}}$, $\mathcal{I}_{\text{fair}}$, and $\mathcal{I}_{\text{chal}}$, respectively.

Since the purpose of this comparison was to find classical NN architectures of which performance was equivalent to that of AlexNet, we increased the number of hidden nodes gradually from 1 to 4. The softmax layer with two nodes was adopted as an output layer as the number of upper arm gym-workouts needed to be classified was two. The rest of the evaluation protocol was the same as introduced in Section 3.1. From Tables 4 and 5, we

can see that the training loss of NNs is smaller than AlexNet. The mean performance of NN almost becomes equivalent or surpasses that of AlexNet when the number of hidden node is set to 4.

**Table 4.** The mean of training loss for AlexNet and neural networks (NNs). h1 through h4 represent the number of hidden nodes in a hidden layer.

| Training Loss | AAFT(1) [†] | | | AAFT(2) | | | AAFT(3) | | | AAFT(4) | | | AAFT(5) | | |
|---|---|---|---|---|---|---|---|---|---|---|---|---|---|---|---|
| | $\mathcal{I}_{easy}$ | $\mathcal{I}_{fair}$ | $\mathcal{I}_{chal}$ | $\mathcal{I}_{easy}$ | $\mathcal{I}_{fair}$ | $\mathcal{I}_{chal}$ | $\mathcal{I}_{easy}$ | $\mathcal{I}_{fair}$ | $\mathcal{I}_{chal}$ | $\mathcal{I}_{easy}$ | $\mathcal{I}_{fair}$ | $\mathcal{I}_{chal}$ | $\mathcal{I}_{easy}$ | $\mathcal{I}_{fair}$ | $\mathcal{I}_{chal}$ |
| AlexNet | 3.37 | 3.28 | 3.10 | 3.07 | 3.13 | 3.27 | 2.55 | 2.17 | 2.95 | 2.11 | 1.82 | 2.42 | 1.34 | 1.18 | 1.57 |
| NN-h1 | 0.20 | 0.20 | 0.26 | 0.20 | 0.22 | 0.21 | 0.23 | 0.21 | 0.15 | 0.25 | 0.20 | 0.20 | 0.25 | 0.25 | 0.21 |
| NN-h2 | 0.07 | 0.20 | 0.25 | 0.18 | 0.19 | 0.25 | 0.15 | 0.02 | 0.25 | 0.20 | 0.15 | 0.26 | 0.20 | 0.20 | 0.24 |
| NN-h3 | 0.10 | 0.10 | 0.15 | 0.15 | 0.15 | 0.15 | 0.15 | 0.03 | 0.25 | **0.01** | 0.15 | 0.26 | **0.15** | 0.21 | **0.15** |
| NN-h4 | **0.03** | 0.16 | 0.24 | **0.06** | 0.15 | 0.24 | **0.01** | **0.01** | 0.24 | **0.01** | **0.01** | 0.15 | **0.15** | **0.15** | **0.15** |

†: AAFT(1) corresponds to no data augmentation.

**Table 5.** The mean of test accuracy for AlexNet and NNs (%). h1 through h4 represent the number of hidden nodes in a hidden layer.

| Test Accuracy | AAFT(1) [†] | | | AAFT(2) | | | AAFT(3) | | | AAFT(4) | | | AAFT(5) | | |
|---|---|---|---|---|---|---|---|---|---|---|---|---|---|---|---|
| | $\mathcal{I}_{easy}$ | $\mathcal{I}_{fair}$ | $\mathcal{I}_{chal}$ | $\mathcal{I}_{easy}$ | $\mathcal{I}_{fair}$ | $\mathcal{I}_{chal}$ | $\mathcal{I}_{easy}$ | $\mathcal{I}_{fair}$ | $\mathcal{I}_{chal}$ | $\mathcal{I}_{easy}$ | $\mathcal{I}_{fair}$ | $\mathcal{I}_{chal}$ | $\mathcal{I}_{easy}$ | $\mathcal{I}_{fair}$ | $\mathcal{I}_{chal}$ |
| AlexNet | 88.4 | 83.3 | 52.1 | **96.3** | 77.5 | 63.5 | 98.5 | 99.3 | 68.9 | 99.6 | **99.7** | 76.6 | 99.7 | **99.9** | 95.8 |
| NN-h1 | 65.4 | 64.9 | 76.2 | 79.1 | 67.2 | 60.0 | 68.3 | 63.5 | 55.0 | 79.4 | 88.3 | 60.0 | 79.4 | 69.8 | 60.0 |
| NN-h2 | 92.4 | 86.1 | 80.0 | 83.9 | 84.6 | 70.0 | 93.0 | 98.1 | 70.0 | 89.2 | 94.1 | 75.0 | 89.3 | 89.9 | 85.0 |
| NN-h3 | 91.3 | 90.8 | 95.0 | 89.9 | 91.8 | 95.0 | 94.0 | 98.4 | 80.0 | 99.1 | 93.9 | 75.0 | 94.6 | 89.9 | 95.0 |
| NN-h4 | **98.4** | 88.5 | 85.0 | 95.5 | 91.8 | 85.0 | **99.6** | 99.4 | 84.2 | 99.3 | 99.6 | 95.0 | 94.8 | 94.6 | 95.0 |

†: AAFT(1) corresponds to no data augmentation.

Tables 6 and 7 show the comparison result for AlexNet versus other deep neural network architectures. Interestingly, VGG-19 reported the best performance in terms of both training loss and test accuracy. For our upper arm gym-workout classification, the performances of ResNet-50 and Inception-v4 were lower than the other two architectures. Further discussion about all the comparison results will be followed in the next section.

**Table 6.** The mean of training loss for AlexNet, VGG-19, ResNet-50, and Inception-v4.

| Training Loss | AAFT(1) [†] | | | AAFT(2) | | | AAFT(3) | | | AAFT(4) | | | AAFT(5) | | |
|---|---|---|---|---|---|---|---|---|---|---|---|---|---|---|---|
| | $\mathcal{I}_{easy}$ | $\mathcal{I}_{fair}$ | $\mathcal{I}_{chal}$ | $\mathcal{I}_{easy}$ | $\mathcal{I}_{fair}$ | $\mathcal{I}_{chal}$ | $\mathcal{I}_{easy}$ | $\mathcal{I}_{fair}$ | $\mathcal{I}_{chal}$ | $\mathcal{I}_{easy}$ | $\mathcal{I}_{fair}$ | $\mathcal{I}_{chal}$ | $\mathcal{I}_{easy}$ | $\mathcal{I}_{fair}$ | $\mathcal{I}_{chal}$ |
| AlexNet | 3.37 | 3.28 | 3.10 | 3.07 | 3.13 | 3.27 | 2.55 | 2.17 | 2.95 | 2.11 | 1.82 | 2.42 | 1.34 | 1.18 | 1.57 |
| VGG-19 | 0.68 | **0.64** | 0.69 | 0.63 | **0.62** | 0.66 | 0.12 | **0.01** | 0.29 | **0.00** | **0.00** | **0.00** | **0.00** | **0.00** | **0.00** |
| ResNet-50 | 1.18 | 1.77 | 1.34 | 2.39 | 2.11 | 2.96 | 4.67 | 6.02 | 4.33 | 4.33 | 3.25 | 5.15 | 4.77 | 3.87 | 4.14 |
| Inception-v4 | 53.95 | 59.80 | 53.70 | 15.37 | 14.80 | 15.65 | 24.83 | 16.98 | 7.26 | 1.83 | 5.37 | 0.99 | 0.13 | 0.45 | 1.54 |

†: AAFT(1) corresponds to no data augmentation.

**Table 7.** The mean of test accuracy for AlexNet, VGG-19, ResNet-50, and Inception-v4 (%).

| Test Accuracy | AAFT(1) [†] | | | AAFT(2) | | | AAFT(3) | | | AAFT(4) | | | AAFT(5) | | |
|---|---|---|---|---|---|---|---|---|---|---|---|---|---|---|---|
| | $\mathcal{I}_{easy}$ | $\mathcal{I}_{fair}$ | $\mathcal{I}_{chal}$ | $\mathcal{I}_{easy}$ | $\mathcal{I}_{fair}$ | $\mathcal{I}_{chal}$ | $\mathcal{I}_{easy}$ | $\mathcal{I}_{fair}$ | $\mathcal{I}_{chal}$ | $\mathcal{I}_{easy}$ | $\mathcal{I}_{fair}$ | $\mathcal{I}_{chal}$ | $\mathcal{I}_{easy}$ | $\mathcal{I}_{fair}$ | $\mathcal{I}_{chal}$ |
| AlexNet | 88.4 | 83.3 | 52.1 | 96.3 | 77.5 | 63.5 | 98.5 | 99.3 | 68.9 | 99.6 | 99.7 | 76.6 | 99.7 | 99.9 | 95.8 |
| VGG-19 | **94.2** | 66.7 | 76.3 | 98.5 | **98.8** | 97.9 | **100.0** | 99.9 | 99.0 | **100.0** | **100.0** | **100.0** | **100.0** | **100.0** | **100.0** |
| ResNet-50 | 50.0 | 40.0 | 50.0 | 50.0 | 50.0 | 50.0 | 50.0 | 50.0 | 50.0 | 50.0 | 50.0 | 50.0 | 51.0 | 50.0 | 51.0 |
| Inception-v4 | 55.0 | 43.3 | 50.0 | 53.7 | 58.8 | 50.0 | 54.2 | 51.3 | 58.3 | 80.7 | 70.8 | 75.3 | 95.7 | 91.7 | 86.8 |

†: AAFT(1) corresponds to no data augmentation.

## 4. Discussion

Recall that the main idea of our approach is to merge joint kinematic data into sEMG data and visualize as a heatmap. We wanted for the employed CNN (AlexNet) to impute a relationship between muscle activation and joint movement via CNN and solve an upper arm gym-workout classification problem. The novelty of the proposed approach is to

control the level of manipulation for the recognizability of input image features as well as the level of data augmentation by AAFT. We wanted to reveal the effect of those two control factors on CNN training loss and test accuracy by statistical analysis. Furthermore, finding the optimal combination of the two factors for the best CNN performance was a part of our research questions.

Our idea of visualizing muscle activation together with joint movement as a heatmap indeed is initiated by approaches to demystify the functional connectivity of a brain across different regions [55,56]. However, existing research has been reported that it is not so straightforward to define biopotential-kinematic relationship explicitly [57–59]; hence, we want to count on CNN to impute those relationship implicitly. According to the statistical analysis, the two control factors had significant main effects for both training loss and test accuracy. Therefore, this finding could be the answers to our first and second research questions.

Table 1 shows that $\mathcal{I}_{\text{fair}}$ has a tendency of having a less value (which means better in terms of training loss) than $\mathcal{I}_{\text{easy}}$ in overall. However, $\mathcal{I}_{\text{easy}}$ has the least standard deviation than $\mathcal{I}_{\text{fair}}$ except AAFT(1). This implies that $\mathcal{I}_{\text{easy}}$ might contribute to the consistency of CNN training. From Table 2, furthermore, $\mathcal{I}_{\text{easy}}$ shows the best test accuracy compared to the others under AAFT(1) and AAFT(2). This can be interpreted as $\mathcal{I}_{\text{easy}}$ is the best suitable choice when data augmentation is not considered. The effect of data augmentation becomes dominant as the level of AAFT increases; accordingly, the test accuracy for $\mathcal{I}_{\text{easy}}$ and $\mathcal{I}_{\text{fair}}$ almost reached to 99% approximately even after AAFT(3). $\mathcal{I}_{\text{chal}}$ shows the monotonically increasing test accuracy as the level of AAFT increases. Especially, the test accuracy of $\mathcal{I}_{\text{chal}}$ increases drastically from 76.6 under AAFT(4) to 95.8 under AAFT(5). This monotonically increasing characteristic can be advantageous over the others for determining design parameters when we consider more complicated motion with the large enough data samples.

The best test accuracy shown by $\mathcal{I}_{\text{easy}}$ for AAFT(1) and AAFT(2) might be interpreted as if the relationship between muscle activation and joint movement was somehow successfully imputed via CNN. Additionally, the statistical analysis indicated that the test accuracy of both $\mathcal{I}_{\text{easy}}$ and $\mathcal{I}_{\text{fair}}$ is significantly higher than that of $\mathcal{I}_{\text{chal}}$ upto AAFT(4). Nevertheless, pairwise comparison could not find any significant difference between $\mathcal{I}_{\text{easy}}$ and $\mathcal{I}_{\text{fair}}$ in terms of the performance metrics of CNN. This might be caused by either a lack of the number of gym-workouts to be classified or a way more dexterous and stronger automatic feature extraction of CNN. These issues can be addressed by increasing the number of gym-workouts to be classified and investigating the sparsity and entropy of $\mathcal{I}_{\text{easy}}$ and $\mathcal{I}_{\text{fair}}$ using methods presented in [60]. In addition, the interaction effect of the two control factors indicate that the optimal combination of a representation method for biopotential-kinematic data together with the level of data augmentation must be involved as a design parameter for an exercise monitoring system.

For the comparison result, in Table 4, the NNs presents the less training loss than that of AlexNet due to a simpler architecture. From Table 5, we can see that the NNs outperforms AlexNets if the number of hidden nodes is greater than 2; however, further evaluation and investigation is needed for the case of the larger class of upper arm gym-workouts. As a result of comparison among deep neural networks presented in Tables 6 and 7, VGG-19 shows the best performance. This result could be expected since VGG-19 consists of the enhanced convolution and pooling layers compared to AlexNet. For both AlexNet and VGG-19, $\mathcal{I}_{\text{easy}}$ yields the best performance under AAFT(1). The training of Inception-v4 improves drastically as the level of AAFT increases and its test accuracy reaches to 90% approximately under AAFT(5). This indicates that Inception-v4 requires more number of data compared to AlexNet and VGG-19. According to Table 7, the training of ResNet-50 might be prematured due to a lack of number of data samples.

## 5. Conclusions

Throughout this paper, we proposed an approach to upper arm gym-workout classification problem according to the spatio-temporal features of sEMG and joint kinematic data. First, in our approach, two upper arm gym-workouts—dumbbell curl (target muscle: biceps brachii) and dumbbell kickback (target muscle: triceps brachii), respectively—were demonstrated by a professional trainer. During the demonstration, sEMG data sample and elbow joint angle data sample were measured and stored by Myo armband and Kinect v2 at every 20 ms, respectively. Next, after RMS filtering and integrating, the processed both data were merged into and visualized as one image with considering the level of manipulation for the recognizability of input image features from human eyes' point of view. Finally, CNN (AlexNet) was employed to impute a relationship between muscle activation and joint kinematics by being trained with the visualized image data set as well as to solve the gym-workout classification problem.

The statistical result on CNN performance metrics showed that our approach—controlling the level of manipulation for the recognizability—had a significant main effect on CNN performance; of course, so did the level of data augmentation. We also found that there were the interaction effects of two control factors, which should be considered to find the optimal combination of the level of data augmentation and the level of manipulation for the recognizability of input image features as design parameter. Pairwise comparison did not reveal any significant difference when an input image data was visualized as $\mathcal{I}_{\text{easy}}$ or $\mathcal{I}_{\text{fair}}$. However, both visualization approaches showed the outperformance over $\mathcal{I}_{\text{chal}}$.

The contribution and innovation by disseminating our findings can be summarized as follows. First, this study proposes a novel approach to approximate a relationship between muscle physiology and joint kinematics via CNN feature extraction. Second, by providing a systemic procedure as well as a quantitative analysis, this study advances the state of the art in the problem of developing exercise monitoring systems. Finally, the outcomes related to the level of AAFT and the visualization technique can be utilized to determine design parameters when we develop the exercise monitoring system which is compact and cost-affordable.

Future studies should be followed in the directions of increasing the number of gym-workouts to be classified as well as investigating the sparsity and the entropy of the input image data according to the level of manipulation for the recognizability. Based on findings in this study, the proposed approach will serve as a core and be culminated to the development of an exercise monitoring system by which trainee's muscular physiology as well as joint kinematics are properly monitored.

**Author Contributions:** Conceptualization, J.-H.Y. and H.-U.Y.; methodology, J.-H.Y. and H.-U.Y.; software, J.-H.Y., H.-J.J., Y.-S.J., Y.-B.K., C.-J.L., S.-T.S. and H.-U.Y.; validation, H.-J.J. and H.-U.Y.; formal analysis, J.-H.Y., H.-J.J., C.-J.L., S.-T.S. and H.-U.Y.; investigation, H.-U.Y.; resources, J.-H.Y., H.-J.J., C.-J.L. and H.-U.Y.; data curation, J.-H.Y., H.-J.J., Y.-S.J., Y.-B.K. and H.-U.Y.; writing—original draft preparation, J.-H.Y., H.-J.J., Y.-S.J., Y.-B.K., C.-J.L. and H.-U.Y.; writing—review and editing, J.-H.Y., H.-J.J., S.-T.S. and H.-U.Y.; visualization, J.-H.Y., H.-J.J. and H.-U.Y.; supervision, H.-U.Y.; project administration, H.-U.Y.; funding acquisition, H.-U.Y. All authors have read and agreed to the published version of the manuscript.

**Funding:** This research was supported by the Ministry of Science and ICT, Korea, under the National Program for "Excellence in SW (Grant Number: 2019-0-01219)" supervised by the Institute of Information and Communications Technology Planning and evaluation (IITP).

**Conflicts of Interest:** The authors declare no conflict of interest.

**Abbreviations**

The following abbreviations are used in this manuscript:

| | |
|---|---|
| sEMG | Surface electromyography |
| RNN | Hybrid soft actuator module |
| LSTM | Long shot-term memory |
| DBN | Deep belief network |
| DNN | Deep neural network |
| NN | Neural network |
| IMU | Inertial measurement unit |
| CNN | Convolutional neural network |
| PS | Pilot subject |
| PD | Protocol director |
| AAFT | Amplitude adjusted Fourier transform |

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
