# Peer review of "Classifying Upper Arm Gym-Workouts via Convolutional Neural Network by Imputing a Biopotential-Kinematic Relationship"

_applsci, doi:10.3390/app11062845_

Round 1

Reviewer 1 Report

This is an interesting research paper. There are some suggestions for revision. 

1. The motivation is not clear. Please specify the importance of the proposed solution. 

2. Please highlight your contributions in introduction. 

3. As shown in Section 2.2, the AAFT related details are from existing work. Please specify your contributions.

4. Please explain the relationship between L_easy/L_fair/L_chal and the corresponding Eq. 1/2/3 and 4. 

5. Section 2.4 is too weak. More analysis details should be discussed. 

6. The experimental results are not convincing. Please compare the proposed solution with existing solutions. 

7. It seems the proposed solution is a combination of several existing solutions. Please specify your contributions and innovations. 

Author Response

Please refer to the attached authors' response. Thanks for all your valuable comments.

Reviewer 2 Report

This manuscript presents a framework for upper arm gym-workout classification based on both biopotential and joint kinematics data. The joint kinematic data and multichannel electromyography signal are augmented and visualized as one image and classified by a CNN. This framework achieves an accuracy of more than 90%. Effects of different data augmentation levels and manipulation levels for the recognizability of input image features were also investigated.

Major comments:

  1. It’s better to compare the CNN classification results with traditional feature extraction and classification method, to show how much contribution the CNN framework makes in the classification step.
  2. Some evaluation details (in Chapter 3) are missing, e.g., the size of the training / testing data set; details of the CNN settings, etc.
  1. In Figure 6, the description wrote: “No significant difference was found between Ieasy and Ifair across all AAFT levels.”, which seems not consistent with the mark in Figure 6(b)?
  2. Experiment results show that both Ieasy and Ifair outperform Ichal. And Ichal has a higher computation complexity. Please add some discussion of the advantage and possible application of Ichal.

Author Response

Please refer to the attached author's response. Thanks for all your valuable comments.

Round 2

Reviewer 1 Report

All my concerns have been addressed. I recommend this paper for publication. 

Author Response

All authors sincerely thank you for your comments during the reviewing process.

Reviewer 2 Report

This manuscript presents a framework for upper arm gym-workout classification based on both biopotential and joint kinematics data. The joint kinematic data and multichannel electromyography signal are augmented and visualized as one image and classified by a CNN. This framework achieves an accuracy of more than 90%. Effects of different data augmentation levels and manipulation levels for the recognizability of input image features were also investigated.

Minor comments:

The authors should double check the manuscript to avoid grammatical errors and typos, e.g.,

Line 382: The contribution and innovation by disseminating our findings can by summarized as follows. -> can be summarized

Author Response

Grammatical errors and typos have been clarified by a strong proof-reading process.

All authors sincerely thank you for your comments during the reviewing process.